# Host Resistance, Genomics and Population Dynamics in a *Salmonella* Enteritidis and Phage System

**DOI:** 10.3390/v11020188

**Published:** 2019-02-22

**Authors:** Angela Victoria Holguín, Pablo Cárdenas, Catalina Prada-Peñaranda, Laura Rabelo Leite, Camila Buitrago, Viviana Clavijo, Guilherme Oliveira, Pimlapas Leekitcharoenphon, Frank Møller Aarestrup, Martha J. Vives

**Affiliations:** 1Department of Biological Sciences, Universidad de Los Andes, 111711 Bogotá, Colombia; p.cardenas10@uniandes.edu.co (P.C.); cat-prad@uniandes.edu.co (C.P.-P.); mc.buitrago10@uniandes.edu.co (C.B.); in-clavi@uniandes.edu.co (V.C.); mvives@uniandes.edu.co (M.J.V.); 2Instituto René Rachou, Fundação Oswaldo Cruz, 21040-900 Belo Horizonte, Brazil; rabelo.leite@gmail.com (L.R.L.); guilherme.oliveira@itv.org (G.O.); 3Instituto Tecnológico Vale, 66055-090 Belém, Brazil; 4Research Group for Genomic Epidemiology, National Food Institute, Technical University of Denmark, 2800 Kgs. Lyngby, Denmark; pile@food.dtu.dk (P.L.); fmaa@food.dtu.dk (F.M.A.)

**Keywords:** phage-therapy, *Salmonella* Enteritidis, bacteria-phage coevolution, antibiotics, bacterial resistance

## Abstract

Bacteriophages represent an alternative solution to control bacterial infections. When interacting, bacteria and phage can evolve, and this relationship is described as antagonistic coevolution, a pattern that does not fit all models. In this work, the model consisted of a microcosm of *Salmonella enterica* serovar Enteritidis and *φ*San23 phage. Samples were taken for 12 days every 48 h. Bacteria and phage samples were collected; and isolated bacteria from each time point were challenged against phages from previous, contemporary, and subsequent time points. The phage plaque tests, with the genomics analyses, showed a mutational asymmetry dynamic in favor of the bacteria instead of antagonistic coevolution. This is important for future phage-therapy applications, so we decided to explore the population dynamics of *Salmonella* under different conditions: pressure of one phage, a combination of phages, and phages plus an antibiotic. The data from cultures with single and multiple phages, and antibiotics, were used to create a mathematical model exploring population and resistance dynamics of *Salmonella* under these treatments, suggesting a nonlethal, growth-inhibiting antibiotic may decrease resistance to phage-therapy cocktails. These data provide a deep insight into bacterial dynamics under different conditions and serve as additional criteria to select phages and antibiotics for phage-therapy.

## 1. Introduction

So-called “superbugs,” or bacteria resistant to multiple antibiotics, are a worldwide problem that urgently requires solutions in order to control bacterial infections. Bacteriophages (henceforth “phages”) have been proposed as one of these options since their discovery. Phages were used as a therapy for a few years until penicillin was discovered. Following the advent of antibiotics, Western countries halted research on phages, and antibiotics were used widely until a few years ago when reports on bacterial resistance increased, and the world started to look for alternatives [1]. Subsequently, phages emerged once again, and several studies have been conducted to understand phage–bacteria interaction and the applicability of these in food, animals, and humans [1].

*Salmonella* was chosen as a model due to the problem it represents for public health in the world and other countries. This bacterium causes salmonellosis, a form of food poisoning in humans. *S. enteritidis*, particularly, can live in chickens without causing disease and pass through them to humans [2]. Between 2008 and 2013, the incidence of *Salmonella* infection in the United States was 15.2 illnesses per 100,000 individuals [2]; 1.2 billion illness and 450 deaths were reported to occur due to non-typhoidal *Salmonella* in the United States, based on the annual report by the Centers for Disease Control and Prevention (CDC) [3]. Globally, strains of *Salmonella* sp. have been found in poultry farms at the main productions zones [1], and they have shown resistance to antibiotics allowed for industrial use, leaving farmers with little alternatives to control it.

Several phage and antibiotic resistance mechanisms have been described for *Salmonella*, including clustered regularly interspaced short palindromic repeats-Cas (CRISPR-Cas) systems for phages, and efflux pumps, antibiotic-altering, and antibiotic-degrading enzymes, for antibiotics; target modification and abortive infection are mechanisms that bacteria use both for phages and antibiotics. Target modification occurs when a mutation alters the phage binding receptor. This mechanism can be carried out in different ways, such as direct modification of the receptor protein, production of exopolysaccharide, or masking the protein [4]. In the direct modification of the receptor, the bacterium changes the active site of the protein or the structural domains involved. The latter option is the most common, because these domains present non-conserved regions, unlike the active site of the protein [5]. On the other hand, the exopolysaccharide produced around the receptor constitutes a physical barrier that the bacteria develops to prevent binding [5]. Finally, the bacterium can synthesize a protein that masks the receptor thereby preventing the phage binding to the bacteria [4].

Bacterial receptors may limit phages to parasite them. Peptidoglycans are essential for bacterial growth and have distinct structures from eukaryotic cells; the different compositional combinations can act as phage receptors. Among the peptidoglycans are transpeptidases, β-lactamases, transglicosilases, and glycopeptides, which also act as targets for antibiotics and are related to antibiotic resistance. Mutations that decrease susceptibility to phages have been found in these proteins [6]. In some cases, bacteria need to make another series of changes to compensate for modifications occurring on the cell wall [7]. The PmrA/PmrB system has been reported in *Salmonella* as a target for a resistance modification mechanism allowing the bacteria to avoid the antibiotic attack, phage infections, and even detection by host immune systems. Briefly, this system is found in the lipid A region of lipopolysaccharides, detects binding to the receptor, and emits a signal which activates the synthesis of a 4-aminoarabinose group that binds to the phosphate ester bound in the diglucosamine component of lipid A, thus decreasing the affinity to the antimicrobial component [6,8].

Another mechanism that bacteria use to fight both antibiotics and phages, also reported in *Salmonella*, are abortive infections (Abi) systems. Abi refers to a collection of bacterial defense systems intended to regulate genes involved in biofilm formation functions, control of bacterial growth (for example, bacteriostatic toxins), and programmed cell death. One of these is the toxin-antitoxin system, which works based on a complex of two proteins. In general, the antitoxin, as its name indicates, works by blocking the toxin, thus preventing cell death. This state is preserved when conditions are appropriate for the bacteria. Once the alarm system is turned on (an external switch), a signal is emitted, and separation of the complex occurs, releasing the toxin and causing cell death [9,10].

Finally, another important mechanism of bacterial resistance to phage is the CRISPR system. This is a system of two clusters, a CRISPR cluster, and a protein cluster, considered by many a form of a bacterial adaptive immune system. Cas proteins are the most studied variation of CRISPR systems, which is why it is usual to find the system named CRISPR-Cas [11].

Given that phage therapy is considered an alternative for the treatment of antibiotic-resistant bacteria, understanding the evolutionary dynamics between *Salmonella* and their phages is important to avoid rapid exhaustion of the therapy, as already happened with antibiotics. Buckling and Rainey [12] described antagonist coevolution in a *Pseudomonas fluorescens*-phage model. In contrast, Barbosa et al. [13] described that a non-coevolutionary interaction took place between *Vibrio* sp. and three different individual phages; instead, the bacteria showed continued resistance in the early stages of the experiment. Based on similar results in *Escherichia coli*, Lenski [14] had previously proposed a mutational asymmetry as an alternative hypothesis for the bacteria–phage interaction. This suggests that when bacteria and phage coexist, bacteria have an advantage over phages because it is easier for the bacteria to mutate the phage receptor than for the phage to find the exact mutation for the new receptor. This sort of mutational asymmetry in favor of the host bacteria has been observed in multiple other bacteria–phage systems as well [14,15,16,17,18,19]. With the aim of understanding the evolutionary interaction between phages and its host, we characterized in vitro and in silico the interaction of previously isolated bacteriophages with a *Salmonella* strain, using different conditions such as a single phage, multiple phages, and antibiotics plus phage. For the phage experiments, we used a modified experimental co-evolution model previously reported and compared the results of this bacterial model with different ones to gain insight into the bacterial response to phage pressure. Here, we used a strain of *Salmonella enterica* subspecies *enterica* sv. Enteritidis (from now: *S.* Enteritidis), and many Colombian native phages that infect the strain.

## 2. Materials and Methods

### 2.1. Microorganisms

Ten strains of *S. enterica* subspecies *enterica* sv. Enteritidis and eight strains of *S. enterica* subspecies *enterica* sv. Typhimurium were previously isolated and characterized by the Colombian Integrated Program for Antimicrobial Resistance Surveillance (COIPARS) group, from the Colombian Corporation for Agricultural Research (*Corporación Colombiana de Investigación Agropecuaria*, CORPOICA). The strains were kindly provided by Pilar Donado, head of the COIPARS group. *Salmonella* Enteritidis strain s25pp was selected for subsequent experiments based on previous studies that showed it was an efficient host to propagate different phages, its rapid growth, and absence of prophage induction by ultraviolet light [20].

### 2.2. Phages

Three *S.* Enteritidis phages previously isolated and characterized, *φ*San23, *φ*San24, and *φ*San15, [21] were selected for their diverse host range and infection parameters [20]. Phage suspension titer was standardized by serial dilution on double agar plaque assays as described by Kropinski et al. [22]. Chloramphenicol was chosen for the evolution experiments with antibiotic due to the presence of a functional AcrAB-TolC efflux pump system in the *S.* Enteritidis genome sequenced in this study, as revealed by preliminary sequence analysis. The AcrAB-TolC efflux pump system found in different Enterobacteriaceae can transport multiple antibiotics, including chloramphenicol, across both the inner and outer membrane, and have been found to be involved in resistance to these compounds [23,24,25]. *S.* Enteritidis s25pp used in this study was found to be tolerant to low concentrations of chloramphenicol (<0.05 µg/mL), albeit exhibiting decreased growth.

### 2.3. Media

Culture media and components were obtained from Oxoid Ltd (Hampshire, UK).

### 2.4. Lysate Preparation

The lysate phage preparation was performed before the coevolutionary assay in order to have purified, concentrated phage. Purified plaques were resuspended in SM buffer overnight at 4 °C and then centrifuged at 4500 rpm, for 20 min at 4 °C. The supernatant was placed in a new tube, and serial dilutions were performed to 10^−5^, using SM buffer as the diluent. Each dilution was plated onto the solid medium by using a double agar technique: 100 µL of each dilution with 100 µL of an overnight culture of the bacteria were placed together in melted top agar (tubes filled with nutrient broth and 0.4% bacteriological agar) and incubated at 37 °C overnight [22,26]. Top agar was removed the next day from plates with near-confluent lysis, using a glass microscope slide. Plates were rinsed with 3 mL SM buffer, and transferred to a new tube; 0.5 mL of chloroform was added, tubes were vortexed for 1–2 min, and centrifuged for 20 min, 4500 rpm, at 4 °C. Then, the supernatant was placed in a new tube with 100 µL of chloroform. Finally, the concentration of each phage solution was determined using the double agar method, counting the plaque-forming units per milliliter (PFU/mL) [27].

### 2.5. S. Enteritidis and Phage *φ*San23 Evolutionary Dynamics

We performed coevolutionary experiments following the Barbosa et al. methodology [13]. Six microcosms (25 mL Erlenmeyer flasks with 6 mL of Luria–Bertani (LB) broth) were carried out by co-cultivating bacteria and phage, at a multiplicity of infection (MOI) of 10, based on Barbosa’s protocol (1 × 10^7^ PFU/mL and 1 × 10^6^ CFU/mL). The microcosms were incubated at 37 °C for 48 h. After this time, samples were taken in order to isolate 10 bacterial colonies and one phage suspension per microcosm. These samples were called transfer number 1, and 6 µL of each microcosm were transferred to a new flask with fresh media. The second set of microcosms was incubated at 37 °C for 48 h. Again, 10 bacterial colonies and one phage suspension were isolated per microcosm, and 6 µL of each one was transferred to fresh media. This procedure was repeated until transfer number 6. Six replicates of the microcosms were performed. Each bacterial clone from each transfer was challenged against the phage suspensions from two transfers forward (*future* phages), from the same transfer (*contemporary* phages), and from two previous transfers (*past* phages). Additionally, all bacterial clones from each transfer were challenged against the ancestral phage, and the phage suspensions from each transfer were challenged against the ancestral bacteria [13].

Control experiments consisted of the bacterial evolutionary control and the phage evolutionary control. For the bacterial evolutionary control, 60 µL of an overnight bacterial culture (OD600 nm 0.8) were transferred to a 6 mL of fresh LB media and the culture was incubated at 37 °C for 48 h (without phage). After this time, samples were taken to isolate 10 bacterial colonies. These samples were named bacterial control transfer number 1, and 6 µL of culture were transferred to a new flask with fresh media and incubated at 37 °C for 48 h and, again, 10 bacterial colonies were isolated, and 6 µL were transferred to fresh media. This procedure was repeated until bacterial control transfer number 6. Each bacterial clone from each bacterial control transfer was challenged against the original, or ancestral, phage (Appendix A).

Phage evolutionary controls were carried out by co-cultivating bacteria and phage at an MOI of 10 (1 × 10^7^ PFU/mL and 1 × 10^6^ CFU/mL). The microcosms were incubated at 37 °C for 48 h, after which centrifugation was performed (4,500 rpm, for 30 min at 4 °C); 60 µL of supernatant was transferred to a 6 mL of fresh LB media, and 60 µL of fresh overnight bacteria culture (OD600 nm 0.8) was added and incubated at 37 °C for 48 h. After this time, samples were taken to recover the phage suspensions. These samples were named phage control transfer number 1. This procedure was repeated until phage control transfer number 6. Each phage suspension from each phage control transfer was challenged against the ancestral bacteria.

### 2.6. Genomic Analyses

In order to identify the changes in bacteria and phages, whole genome sequencing was performed in the samples taken from the experiment. We sequenced 60 bacterial genomes (10 randomly selected clones per transfer, per six transfers, for a total of 60 bacterial genomes); six phages suspensions (one phage suspension per transfer, for a total of six phage metagenomes); 12 bacterial genomes from the control experiments (four randomly selected clones from transfer 1, four from transfer 3, and four from transfer 6); the ancestral bacterial genome; and the ancestral phage genome.

### 2.7. DNA Extraction

Bacterial DNA was extracted using the Easy-DNATM kit, according to manufacturer’s instructions (InvitrogenTM by Life TechnologiesTM, Frederick, MD 21704, USA). Phage DNA extraction was performed using the phenol-chloroform method from a phage suspension with a titer higher or equal to 1 × 10^11^ PFU·mL^−1^. 700 µL phenol:chloroform:isoamyl alcohol (25:24:1) were added to 700 µL of the phages. This mix was centrifuged at 13,000 rpm for 4 min at 4 °C. The aqueous phase was transferred to a new tube and 350 µL of chloroform and 350 µL of isoamyl alcohol were added to it; the mixture was centrifuged at 13,000 rpm for 4 min at 4 °C. The aqueous phase was taken and transferred to a new Eppendorf tube, and the last step was repeated. The aqueous phase was again transferred to a fresh tube, and a tenth of the volume of sodium acetate was added, as well as twice the volume of 100% ethanol. The tubes were stored at −20 °C overnight. The next day the samples were centrifuged at 13,000 rpm for 30 min at 4 °C, the supernatant was discarded and the pellet was washed with 300 µL of 70% ethanol and centrifuged for 5 min; the supernatant was discarded, and the pellet was dried and resuspended in sixty µL of 10 µm Tris-HCl [11].

Next Generation Sequencing, MiSeq Illumina (sequencing at Innovation Centre at McGill University) was carried out. Bacterial and phage genomic data were studied separately. For both, we performed Single Nucleotide Polymorphism SNP calling, using raw reads genomes, using CSI phylogeny software [28,29]. Then, we aligned the mutants against the ancestral genome to determine differences between them using MAUVE [30,31] and BRIG [32,33]. Based on the results, we selected key genes related to bacterial phage-resistance mechanisms, and we modeled the interaction of candidate receptors and tail fiber proteins using DockingServer [34,35].

Bacterial bioinformatics analyses started with a quality assessment using FastQC [36], followed by sequence trimming with Trimmomatic [37,38]. Genomic assembly was performed using different software tools to complete the best assembly, including Assembler 1.0 [39,40], SPAdes [41,42], and the read aligner Bowtie2 [43,44]. The ancestral bacterial assembly was carried first using SPAdes and Assembler 1.0, we selected best assembly based on N50, then the mutants genome assembly was performed with Bowtie2, SPAdes and Assembler 1.0. Gene prediction and annotation was performed using Glimmer [45,46], RAST [47,48], Blast2go [49,50], and Prodigal [51,52]. Phage bioinformatics analyses started with a quality assessment using [36], followed by sequence trimming with Trimmomatic [37,38]. Genomic assembly was performed using SPAdes software [41,42]. Gene prediction and annotation was performed using RAST [47,48]. Phage lifestyle was determined using PHACTS software [53,54].

### 2.8. BtuB, Colicin, Fhu Receptor Knockout

After the genomics results, three genes were selected to be tested experimentally as candidates for bacterial resistance. *BtuB*, *Colicin* and *Fhu* bacterial receptors found in *S*. Enteritidis were knocked out using TargeTron Gene Knockout System (Sigma Aldrich, TM, St. Louis, MO 63104, USA). After the knockout, each mutant was tested by spot test against wild-type bacteriophage to detect the formation of plaques.

### 2.9. Bacterial Growth at Sublethal Chloramphenicol Concentration

A growth curve was carried out to determine the effect of antibiotic at sublethal concentrations on bacterial growth. Growth curves were determined with *S*. Enteritidis both in the absence and presence of chloramphenicol, at 0.05 µg/mL in the latter case. Briefly, for each growth curve, 10 mL nutrient broth were inoculated with a 1:1000 dilution of overnight culture, grown at 37 °C and 100 rpm agitation, and destructively sampled every hour for 10 h. Optical density was measured in three samples per treatment at a wavelength of 550 nm, and samples were dilution plated to 10^−10^ on nutrient agar to determine growth every hour.

### 2.10. Evolutionary Dynamics of S. Enteritidis with Phages and Antibiotic

A series of evolutionary experiments were carried out to study the population dynamics of *S*. Enteritidis and *φ*San23 under different conditions. Bacterial susceptibility to the ancestral *φ*San23 phage was determined for cultures evolving under different treatment regimes in two experiments in order to better examine evolutionary dynamics between one and two phages, respectively, and the bacterial strain. For the first, spanning eight days, treatments included: *S*. Enteritidis in the absence of phages and antibiotic; *S*. Enteritidis in the presence of *φ*San23; *S*. Enteritidis in the presence of *φ*San23 and *φ*San15; and *S*. Enteritidis in the presence of *φ*San23 and chloramphenicol (0.05 µg/mL). For the second experiment, spanning 10 days, treatments included: *S*. Enteritidis in the absence of phages and antibiotic; *S*. Enteritidis in the presence of *φ*San23; *S*. Enteritidis in the presence of *φ*San23 and *φ*San15; *S*. Enteritidis in the presence of *φ*San23 and *φ*San24; *S*. Enteritidis in the presence of *φ*San23, *φ*San15, and *φ*San24; and finally, *S*. Enteritidis in the presence of *φ*San23 and chloramphenicol (0.05 µg/mL). All experiments were carried out at MOI 10 Evolution experiment procedures were adapted from the work of Barbosa et al. [13]. For each treatment, 10 mL cultures were grown for 24 h in two replicates in nutrient broth at 37 °C and 100 rpm. After 24 h, 10 mL of fresh broth were inoculated with 60 µL of culture for each replicate. A supplement of 0.05 µg/mL chloramphenicol was added to the suitable flasks. Transfers were carried out in this way for every day of each experiment (except on day 4). Cultures were streaked and plated onto nutrient agar before every transfer. The susceptibility to *φ*San23 was evaluated for 10 colonies from each culture using a modified protocol of spot assays on double agar [22]. Briefly, 1 mL cultures of each colony were grown overnight in nutrient broth, after which 100 µL of culture were mixed with 100 µL of soft agar and plated onto nutrient agar. 3 µL of *φ*San23 suspension (10 × 10^9^ PFU/mL) were deposited on the surface of each soft agar surface, and plates were inspected for lysis plaques after 24 h. The fraction resistant to *φ*San23 for each treatment was then calculated as the number of soft agar cultures with no lysis plaques divided by 10.

### 2.11. Mathematical Model: Compartment Model Design

A first-order ordinary differential equation (ODE) compartment model was created to study the population dynamics of *S.* Enteritidis and *φ*San23 under the different exposure conditions to phages and antibiotic. Previous models have been applied to study similar bacterial and phage systems in single phage scenarios [55,56,57,58,59,60,61,62,63,64,65]. The model proposed here, shown schematically in Figure 1, studies the dynamics in a population of bacteria exposed to one or more different phages capable of producing cell lysis. To the knowledge of the authors, it is the first ODE model proposed to model population dynamics in a system with explicit multiple phages.

Equation (1) presents the dynamical system equations describing the compartment model. Bacteria are assumed to replicate following a logistic model of specific growth rate *r* with a fixed carrying capacity *K*, a model commonly used for modeling bacterial growth under limited resources [66,67]. The specific growth rate *r* may be modified by an antibiotic, inhibiting growth by a fraction of *α*. The presence and absence of the antibiotic is denoted by the control variable *a*. Bacteria can be susceptible *(S*) or resistant (*R*) to one or both phages (*V*). Transitions between susceptible and resistant bacteria occur at rates determined by a mutation rate *m* dictating a constant rate for single mutations. Transitioning from susceptibility to both phages to resistance to both phages, for example, requires two independent mutations, and occurs at a rate of *m^2^*. Transitions from susceptibility to both phages to resistance to a single phage require a single mutation and occur at a rate of *m − m^2^* to avoid counting double mutants twice. Bacteria susceptible to a given phage may become infected (*I*) by it at a rate dictated by the phage adsorption rate *β*, and the phage and bacterial concentration, following mass-action law kinetics as done frequently in epidemiological models [68]. Infected bacteria lyse and exit the population after the phage’s latent period *λ*, producing new free phage virions at a rate dependent on the phage burst size *b* with every cell lysis. Free phages exit the population both when they infect bacterial cells and through a constant phage degradation rate *µ*. All compartments are given in terms of concentrations (CFU/mL for bacterial compartments, PFU/mL for phage compartments). The model may be solved as a single, continuous culture experiment, or as a series of transferred cultures such as the ones described in the experimental methods. Simulation of culture transfers can be achieved by multiplying all compartments by a constant dilution factor *v*, equal to the inoculation volume as a fraction of the culture volume, at each transfer time.
(1)$$\frac{dS}{dt}=r(1-a\alpha )\left(1-\frac{N}{k}\right)S+\left(m-m^{2}\right)\left(R_{1}+R_{2}\right)+m^{2}R_{12}-\left(2m-m^{2}\right)S-\beta_{1}SV_{1}-\beta_{2}SV_{2}\;\frac{dR_{1}}{dt}=r(1-a\alpha )\left(1-\frac{N}{k}\right)R_{1}+\left(m-m^{2}\right)\left(R_{12}+S\right)+m^{2}R_{2}-\left(2m-m^{2}\right)R_{1}-\beta_{2}R_{1}V_{2}\;\frac{dR_{2}}{dt}=r(1-a\alpha )\left(1-\frac{N}{k}\right)R_{2}+\left(m-m^{2}\right)\left(R_{12}+S\right)+m^{2}R_{1}-\left(2m-m^{2}\right)R_{2}-\beta_{1}R_{2}V_{1}\;\frac{dR_{12}}{dt}=r(1-a\alpha )\left(1-\frac{N}{k}\right)R_{12}+\left(m-m^{2}\right)\left(R_{1}+R_{2}\right)+m^{2}S-\left(2m-m^{2}\right)R_{12}\;\frac{dI_{1}}{dt}=\beta_{1}V_{1}(R_{2}+S)-\frac{1}{\lambda_{1}}I_{1}\;\frac{dI_{2}}{dt}=\beta_{2}V_{2}(R_{1}+S)-\frac{1}{\lambda_{2}}I_{2}\;\frac{dV_{1}}{dt}=\frac{b_{1}}{\lambda_{1}}I_{1}-\beta_{1}V_{1}(R_{2}+S)-\mu_{1}V_{1}\;\frac{dV_{2}}{dt}=\frac{b_{2}}{\lambda_{2}}I_{2}-\beta_{2}V_{2}(R_{1}+S)-\mu_{2}V_{2}\;N=S+I_{1}+I_{2}+R_{1}+R_{2}+R_{12}$$

In the case of a single-phage situation, the system in Equation (1) can be simplified by taking *V_2_* = 0 and redefining *S = S + R_2_* and *R_1_ = R_1_ + R_12_*, obtaining the system in Equation (2).
(2)$$\frac{dS}{dt}=r(1-a\alpha )\left(1-\frac{N}{k}\right)S+m\left(R_{1}-S\right)-\beta_{1}SV_{1}\;\frac{dR_{1}}{dt}=r(1-a\alpha )\left(1-\frac{N}{k}\right)R_{1}+m\left(S-R_{1}\right)\;\frac{dI_{1}}{dt}=\beta_{1}V_{1}S-\frac{1}{\lambda_{1}}I_{1}\;\frac{dV_{1}}{dt}=\frac{b_{1}}{\lambda_{1}}I_{1}-\beta_{1}V_{1}S-\mu_{1}V_{1}\;N=S+I_{1}+R_{1}$$
As noted by previous authors [59,60,61,69], the population of a specific phage will only grow if the population of bacteria susceptible to the phage exceeds a given threshold, the threshold of proliferation (*N_P_*). This threshold can be computed from the basic reproduction number of the phage ${ℛ}_{0}$, understood as the number of new phages produced by a single phage in a population of bacteria entirely susceptible to the phage [68]. This analysis yields the proliferation thresholds *N_P,1_* and *N_P,2_* for Phages 1 and 2, respectively, shown in Equation (3) in terms of proliferation threshold values for each compartment *S_P_*, *R_1,P_*, and *R_2,P_* (see the analysis in the Appendix A).
(3)$$N_{P,1}=S_{P}+R_{2,P}=\frac{\mu_{1}}{\beta_{1}(b_{1}-1)}\;N_{P,2}=S_{P}+R_{1,P}=\frac{\mu_{2}}{\beta_{2}(b_{2}-1)}$$

Additionally, each phage will only cause the population of bacteria susceptible to them to decrease if the phage population exceeds what is known as the inundation threshold [63,70]. An upper bound for each phage’s inundation threshold up to the first peak of infection for each phage, *V_IB_*, can be calculated as shown in Equation (4) (see the analysis in the Appendix A).
(4)$$V_{IB,1}=\frac{r(1-a\alpha )-m}{\beta_{1}}\;V_{IB,2}=\frac{r(1-a\alpha )-m}{\beta_{2}}$$

Model Parameters and Assumptions. Table 1 summarizes information on the different parameters in the model. Bacterial carrying capacity *K* and intrinsic growth rates *r*, both in the presence and absence of antibiotic, were estimated by fitting growth curve data *D* obtained as previously explained to a logistic function of time *t* (Equation (5)).
(5)$$D(t)=\frac{K}{1+e^{-(\phi +rt)}}$$

Infected bacteria were assumed to consume resources from the environment but not replicate, due to the phage’s hijacking of their cellular machinery. As has been done before [63], it is assumed that all non-infected bacteria in the model possess the same replication rate, regardless of their resistance or susceptibility to phage. The lack of a fitness penalty due to resistance to phage was confirmed experimentally, as no appreciable difference was found in vitro in the growth rate of bacteria resistant to φSan15, φSan23, or [20]. The relative growth penalty caused by sublethal concentrations of antibiotic was calculated from the intrinsic growth rates obtained from non-linear fitting in the presence (*r_a_*) and absence (*r_n_*) of antibiotic, as shown in Equation (6).
(6)$$\alpha =1-\frac{r_{a}}{r_{n}}$$

The model assumes the resistance mechanisms to each phage are independent of each other. In the case of φSan23, resistance is mediated by mutation of a surface receptor as was determined in this work; so far, we have no information about the other phages’ resistance mechanisms. Certain mechanisms of resistance, such as capsules or restriction enzymes, increase resistance to multiple phages [5]. This model does not take these mechanisms into account.

Mutation probability was estimated from reported *Salmonella* molecular evolution rates for C→T transitions [71], and molecular data for *S*. Enteritidis identifying bacterial resistance to *φ*San23 in a specific C→T point mutation on membrane receptor *BtuB*, as was shown in this work. This mutation rate is consistent with previous experimental data from fluctuation tests for similar bacteria–phage systems [73,74]. The probability of reversing this mutation can differ from that of the mutation occurring, as can the probabilities of mutations conferring resistance to different phages. Furthermore, it is known that mutation frequency within a single strain can change in response to stress [75,76,77,78]. Nevertheless, for the sake of simplicity, the probability of mutation was assumed to be constant for all types of mutations, independently of whether the mutation involved acquisition or loss of resistance to one phage or another. Similarly, the free phage decay rates of different phages can vary within a range but were taken as an average of free phage decay rates across many Myoviridae phages [63,64,72]. In both the case of mutation probability and free phage decay rate, the behavior of the model was found to be largely insensitive to changes within the values reported for *Salmonella* and Myoviridae, respectively, in the literature [63,64,72,79].

Phage adsorption overestimates successful adsorption rate, given that not all phage irreversible adsorption events result in successful infection terminating in cell lysis. However, adsorption rates provide a reasonable estimate for successful infection rates in lytic phages [80,81,82]. The model does not take phage superinfection of bacteria into account, the underlying assumptions being that the first phage to infect a bacterial cell is the first to cause lysis and the only one releasing fully-formed progeny and that free phage virions adsorbing onto previously infected bacteria represent a negligible fraction of total free virions. The effect of including superinfection of already infected bacterial compartments into the equations for the rates of change in viral populations was found to be minimal according to numerical simulation with the parameters in Table 1. Burst sizes and latent periods were obtained from experimental measurements of *φ*San23 and *φ*San15 one-step curves for Phage 1 and 2, respectively [20]. Adsorption rates were estimated from experimental measurements of *φ*San23 growth curves, with *φ*San15 showing a decreased adsorption rate [20,80].

Implementation and Numerical Solution. Non-linear fitting of growth data was carried out using the Levenberg-Marquardt algorithm for least-squares minimization [83]. The model was solved numerically using the Livermore Solver for Ordinary Differential Equations (LSODA) solver-switching algorithm [84]. In the case of both algorithms, FORTRAN implementations running underneath the wrapper function in R [85] were used. The source code for model simulation and graphics is available online [86].

Accession Numbers: MUNA00000000: *Salmonella enterica* subsp. *enterica* strain s25pp.

## 3. Results

### 3.1. *S*. Enteritidis and Phage *φ*San23 Evolutionary Dynamics

We did not find evidence of antagonistic co-evolution or any arms race between phage *φ*San23 and *S*. Enteritidis s25pp, as shown in Figure 2. The resistance of the bacteria appeared at the first transfer and this resistance was stable over time. In addition, when bacteria from all transfers were challenged against the ancestral phage, all bacteria showed resistance to the ancestral phage. Phages from each transfer from the control where only phages were allowed to evolve showed to be infective against ancestral bacteria (Figure 3). Bacteria isolated from the control where only bacteria were allowed to evolve showed to be susceptible to the ancestral phage.

### 3.2. Genomic Analyses

The genomic analysis comprises three sets of data: the ancestral bacterial genome, the ancestral phage genome, and the co-evolution assay isolates (60 bacterial genomes and 6 phages metagenomes). The ancestral or wild-type bacterial genome was trimmed and assembled; the whole genome shotgun project has been deposited at the DNA Data Bank of Japan DDBJ/ European Nucleotide Archive ENA/GenBank under the accession MUNA00000000. The version described in this paper is version MUNA01000000. We assembled the genome and aimed to identify key proteins related with bacterial resistance mechanisms. During genome annotation of *S*. Enteritidis s25pp wild-type bacteria (Appendix A), two bacterial and phage related resistance mechanisms were identified.

The phage genome analyses were limited due to the lack of information. It is important to recall that more than 80% of phage genome sequences are unknown. We were able to assemble the genome into 9 contigs, and identify 341 coding DNA sequences (CDS), most of them encoding hypothetical proteins (Appendix A). The lifestyle analysis suggested φSan23 is a virulent phage.

The next step was to find changes in the phage-resistant mutants (isolates from the transfers) compared to the ancestral bacterial genome in order to see what mechanism[s] the bacteria are using to prevent phage infection. After assembly and annotations, mutant and control bacterial genomes were plotted using BRIG software (Figure 4). Hence, we aligned the whole genomes and performed SNP calling. The results showed first, there were no variations in the CRISPR-Cas system; second, the prophage exhibited many SNPs throughout the mutants; and third, most of the SNPs correspond to cell wall and capsule proteins (Appendix A) but without any pattern that could be associated to the transfer (Appendix A), including outer membrane vitamin B12/cobalamine receptor BtuB [87], found in the National Center for biotechnology Information (NCBI) database with accession AIE08100; colicin I receptor; and Fhu (Ferric hydroxamate outer membrane receptor FhuA), involved in ferric hydroxamate uptake [88].

### 3.3. BtuB, Colicin, Fhu Receptor knockout

After gene knockout and phage testing, *BtuB* was shown to be causing the resistance in the bacteria against phage *φ*San23. When spot testing was performed in the knockouts for *Colicin* and *Fhu* receptors, infection was not affected. However, when the *BtuB* knockout was tested, spot assays showed that φSan23 was unable to infect the bacteria.

### 3.4. Assessing Bacterial Growth at Sub-Lethal Chloramphenicol Concentration

The specific growth rate for *S*. Enteritidis with no phage or antibiotic was estimated at *r* = 0.68885/h, based on non-linear fitting of growth curve data, similar to previous measurements in *Salmonella* [89]. Carrying capacity for the *S.* Enteritidis population was estimated as *k* = 1.2888 × 10^8^ CFU/mL from the same fit. Addition of sub-lethal concentrations of antibiotic (0.05 µg/mL chloramphenicol) resulted in a 33.7% reduction in *S*. Enteritidis growth rate (Appendix A).

### 3.5. Evolutionary Dynamics of *S.* Enteritidis with Phages and Antibiotic

Evolution experiments were carried out under different treatments combining phages and antibiotic, as explained before. The results of these experiments can be found in Figure 5. The ODE compartment model described above was solved simulating four different treatments: a single-phage system with no antibiotic (Phage 1 initial MOI of 1, Phage 2 initial MOI of 0, a = 0), a single-phage system with antibiotic (Phage 1 initial MOI of 1, Phage 2 initial MOI of 0, a = 1), a two-phage system with no antibiotic (Phage 1 initial MOI of 1, Phage 2 initial MOI of 0.1, a = 0), and a two-phage system with antibiotic (Phage 1 initial MOI of 1, Phage 2 initial MOI of 0.1, a = 1). In all cases, the model was solved with a population of 12,888 susceptible cells and a MOI of 0.1 for Phage 1 as initial conditions, using a time step of 0.01 h. MOI was changed from 10 to 0.1 as we did not want the bacterial population to be killed at the first cycle, but we wanted the phages to reproduced inside the bacteria and to observe the dynamics with this condition. Figure 5 compares the results of all four treatments in terms of total bacterial population and resistance to phage treatment, while Figure 6 presents the dynamics of each compartment in the model for each treatment. As can be seen in Figure 5_bookmark14, resistance to the phage propagates throughout the population rapidly and irreversibly in the case of a single-phage culture with no antibiotic, a phenomenon observed previously by Barbosa et al. [13]. The addition of antibiotic results in extended susceptibility of *S*. Enteritidis to *φ*San23.

As can be seen in Figure 7, the combination of two phages is predicted by the mathematical model to cause an increase in the time taken by the bacterial culture to develop resistance against phage treatment, as has been reported elsewhere [13]. Furthermore, the addition of a growth-reducing, sub-lethal antibiotic is shown by the model to be an equally effective mechanism of increasing the time to complete resistance. This effect, which confirms the experimental data, is evident both in single- and two-phage treatment, and in continuous culture as well as a culture transfer scenario with periodic bottlenecks.

## 4. Discussion

*φ*San23 is a candidate for biocontrol agent against *S.* Enteritidis. This phage, according to the characterization, was found to be a virulent phage forming clear plaques, possessing a broad host range among the *Salmonella* sp. strains of interest to this study, and exhibiting good stability at 4 °C. Therefore, phage *φ*San23 was chosen for the co-evolutionary models. Our experiments revealed a rapid increase in the frequency of resistant bacteria. Bacteria that evolved resistance during the co-evolution experiment remained resistant against the ancestral phage strains. Data collected from the evolution experiment, in which the only phage evolved and bacteria were removed and replaced every transfer, revealed that the infectivity of individual phages decreased drastically over the first day of assessment. This suggests that, in addition to the rapid evolution of bacterial resistance, phages may have also evolved low infectivity in the co-evolution treatment.

These results differ from what has been reported by Buckling and Rainey [12], where the authors suggested antagonistic coevolution between bacteria and phages. The results with *Salmonella* sp. and *φ*San23 did not show an arms race interaction or antagonistic coevolution. This pattern repeated with another phage against the same *Salmonella* strain. Barbosa et al. [13] performed the same methodology using *Vibrio harveyi* as the model. In this case, they tested three individual phages as well as a cocktail of all three. The results from individual phages did not show antagonistic coevolution, but it was found when phages were tested together. *Pseudomonas fluorescens*, which was used as a model by Buckling and Rainey [12], showed a powerful antagonistic coevolutionary trend. Altogether, the data from these studies suggest that the interaction between phages and their host seems to depend on the bacterial and phage model, at least in controlled lab experiments.

The results led us to sequence bacteria and phage genomes and analyze the changes over time. For the bacterial genomic analyses, three interesting candidates for resistance mechanisms were detected. First, CRISPR-Cas systems were the first hypothesis proposed as an explanation for bacterial resistance. Given CRISPR’s surge to fame as one of the strongest explanations for the molecular mechanisms that bacteria use against foreign DNA, it was plausible to think this was the mechanism conferring bacterial resistance to the phage. Nevertheless, once we performed the alignment of this region with the phage-resistant mutants and ancestral bacteria, no changes throughout the sequence were shown. In a similar fashion, we performed an alignment between these sequences and the phage genome in order to confirm no *φ*San23 DNA was found within the CRISPR spacers. Second, the most notable change (the longest region that is different) in the phage-resistant mutants compared to the ancestral bacteria was located in the prophage region. An alternative hypothesis was that this might serve as a site for *φ*San23 genome integration. However, neither the ancestral bacteria nor any of the resistant mutants were found to have any sequence homologous to *φ*San23. Additionally, the phage genome did not present any gene encoding for a protein related to insertion or the lysogenic cycle. Finally, mutations were found in three membrane proteins: BtuB, Colicin I receptor precursor and Fhu receptor. Protein BtuB, a vitamin 12/cobalamin transporter [87], was previously reported as a phage receptor, and the molecular docking interaction modeling performed showed a possible interaction between the BtuB protein and the annotated tail proteins found in the phage genome (Appendix A). The docking models did not predict any interaction for the other two membrane proteins. Knockout results suggest BtuB protein is the receptor for *φ*San23 and the resistance mechanism the bacteria are using to defend themselves against the phage is a functional mutation in the receptor that binds with the phage [90].

In contrast, phage genome analyses did not show a clear path to overcome the bacterial resistance. Random SNPs were found throughout the genome, and none of them showed a functional change. The mutations found in the gene sequences encoding for tail proteins did not represent functional changes, so it is not a direct defense mechanism against bacterial resistance. It seems that a small proportion of the bacterial population remains sensitive to the phage and this is what allows the phage to survive a relatively long time; however, the bacterial population is always predominant in the microcosm.

There are three interesting lessons from these sets of experiments. Firstly, in the controlled lab experiments of coevolution, there is no general pattern in the interaction between phage and bacteria. This interaction depends on the particular bacterial and phage model. Secondly, in the *S*. Enteritidis model, the bacteria seem to respond to the phage pressure with parsimony: it is less energy-consuming to mutate the phage receptor rather than use the complex machinery of CRISPR-Cas systems. Third, one individual phage does not seem to battle the molecular machinery that *S*. Enteritidis uses to defend itself. Our results strongly suggest that the mutational asymmetry gives an evolutionary advantage to the bacteria against the phage. It is easier for the former to mutate a receptor rather than for the latter to find the exact mutation needed to overcome the functional change in the receptor. Therefore, in the long term after forced contact, continuous co-culture and under a strong selective phage pressure, bacteria will prevail.

The mathematical model here studied predicts the growth-reducing effect of the antibiotic increases the effectiveness of phage treatment since it increases the time taken for resistance to appear. As depicted in Figure 6, this occurs partly because the reduction in growth rate delays the onset of the phages’ peaks of infection, but mainly because it delays the post-peak growth of resistant bacteria. This results in longer times for the total bacterial population to reach carrying capacity and for the fraction of bacteria resistant to a single phage to approach totality, both in single culture and culture transfer settings (Figure 7). The addition of a second, weaker phage in the form of *φ*San15 resulted in no noticeable increase in time to resistance, both in vitro (Figure 5), and in silico as predicted by the model (Figure 7). This may be a consequence of the relatively weaker *φ*San15 when compared to *φ*San23 concerning proliferation and inundation thresholds. The onset of the infection peak for Phage 1 (*φ*San23) overtakes the population rapidly due to its lower proliferation (*N_1,P_*) and inundation thresholds *V_1,P_*, delaying the infection peak for Phage 2 (*φ*San15) for many more hours while the bacterial population recovers from the first phage (Figure 6 and Figure 7). This also explains the momentary reversion of resistance to Phage 1 seen in the two phage systems (Figure 7), caused by the exponential growth of bacteria resistant to Phage 1 but not to Phage 2, followed by the subsequent increase of Phage 2 virions once the population reaches the proliferation threshold for Phage 2 *N_2,P_*, and the population collapse caused once the virions reach the inundation threshold *V_2,P_* (Figure 6). These transient reversion dynamics occur rapidly in culture transfer experiments and are not captured in the sample times shown in Figure 5. The use of phages with equal adsorption rates resulted in qualitatively different behavior, with both phages reaching thresholds simultaneously and resulting in increased time before resistance spread through the population, with no transient reversion [20]. This behavior, however, was highly sensitive to differences in the relative magnitude of the adsorption rates, and in practice may not necessarily be stable due to stochastic effects leading one of the phages to reach the inundation threshold before the other has reached the proliferation threshold, resulting in the same qualitative behavior observed before. Although the experimental data fit was closer to the model in the first experiment than in the second, the qualitative behavior in both cases corresponds to that predicted theoretically. Differences in fit may be due to the system’s sensitivity to small variations in experimental parameters, including stochastic variations in initial conditions and transfer inoculum size and composition, particularly during the early stages of the experiment [20]. These variations amplify differences in time scale behavior downstream both in the model and in vitro but preserve the qualitative dynamics here reported. It may be of some use in the future to model the same system stochastically using a Gillespie simulation in order to sound the effect of probabilistic effects on the system’s dynamics.

The theoretical and experimental findings here presented suggest the value of supplementing phage-therapy with a growth-reducing treatment, such as antibiotics for which resistance is already present in the population, capable of enhancing the phages’ population-reducing effect. As has been shown, antibiotics help to prolong the susceptibility of bacterial populations to a given phage or combination of phages. This implies that bacterial population numbers are kept at low levels due to phage population control for significantly longer periods of time. The theoretical work shown here indicates that this can be explained in terms of the effect that sub-lethal antibiotic concentrations have on bacterial reproduction rate. Furthermore, the use of antibiotics at sub-lethal levels may also contribute to modifying phages’ properties, enhancing their effect. Santos et al. [26] observed increases in phage burst size and decreases in latency in the presence of antibiotics; the model presented here does not take these possibilities into account. Additionally, the results show that the combination of any two phages may not have as significant an effect on the time taken to evolve phage resistance as would be desired. Indeed, the qualitative population dynamics and, therefore, the effectiveness of phage-therapy, seem to be heavily dependent on the specific parameters of the phages involved. While the addition of an additional phage does increase the time to complete resistance, the magnitude of this increase varies wildly depending on phage adsorption and burst size. The multiplicative nature of these effects explains why success with three or more phage cocktails has seen experimental success [13], but the work here suggests that the effects of adding a lethal phage can be significantly augmented by reducing the overall growth rate using a sub-lethal antibiotic. These insights may prove valuable in the design of future phage-based alternatives for antibacterial treatment.

The current study showed the phage interaction of a resistant-to-antibiotics *Salmonella* Enteritidis and the dynamics under different treatments. The results determined that the phage–bacteria interaction change from system to system, and this particular interaction did not demonstrate antagonist coevolution but mutational asymmetry. On the other hand, one the most important limitations of the study is that in the real world environment the conditions are different from the conditions we tested and the bacteria may not be that “comfortable” as it was under our conditions.

## Figures and Tables

**Figure 1 viruses-11-00188-f001:**
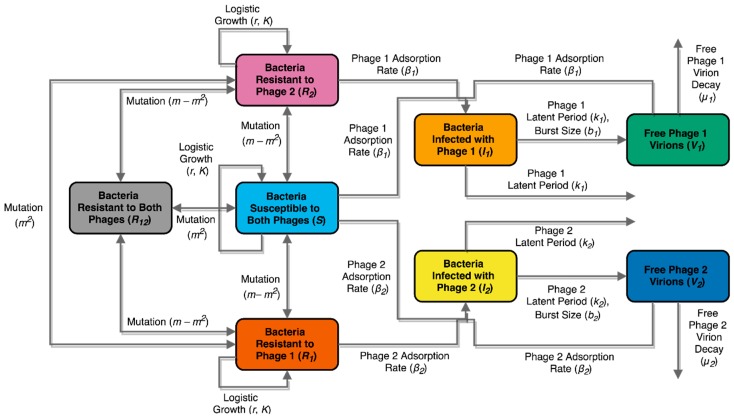
Population compartment model describing dynamics of two distinct phages infecting a population of bacteria. Colored compartments represent subdivisions of the population, corresponding to state variables in the ordinary differential equation (ODE) model. Bacteria may be susceptible to both phages (blue), resistant to one phage and susceptible to the other (light orange), resistant to both phages (dark orange), or infected with one phage (pink). Free phages are denoted in green compartments. Arrows represent population flow between compartments, with constants governing the rate annotated on each.

**Figure 2 viruses-11-00188-f002:**
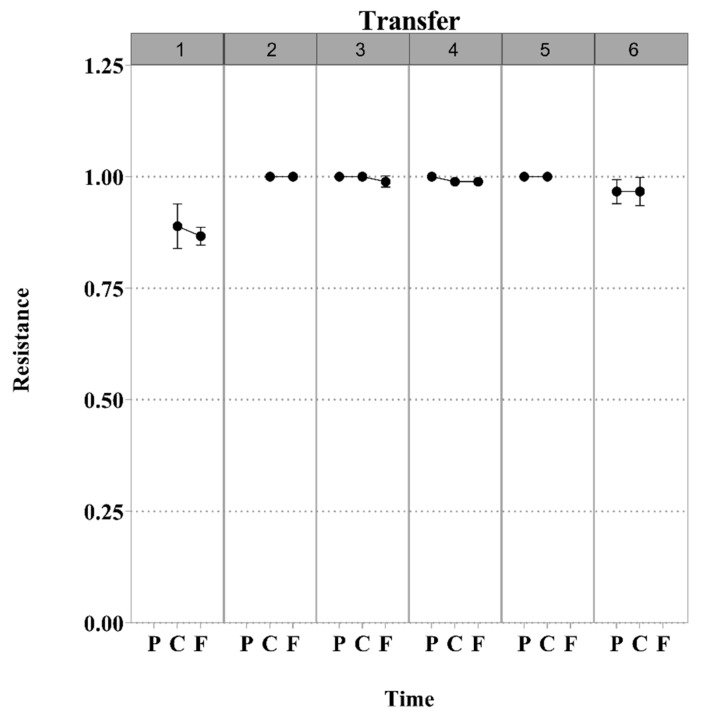
Interaction between *S.* Enteritidis s25pp and phage *φ*San23 under controlled conditions. The lower horizontal axis denotes times P (present), C (Contemporary), and F (Future). The upper horizontal axis shows the transfer number. The vertical axis indicates population resistance frequency, on a scale from 0 to 1.00. The plots indicated that no antagonistic co-evolutionary relationship was established. Phages from each transfer isolated from the control where only the phage was able to evolve were shown to be infective against ancestral bacteria, and bacteria isolated from the control where only bacteria was able to evolve were shown to be susceptible to the ancestral phage.

**Figure 3 viruses-11-00188-f003:**
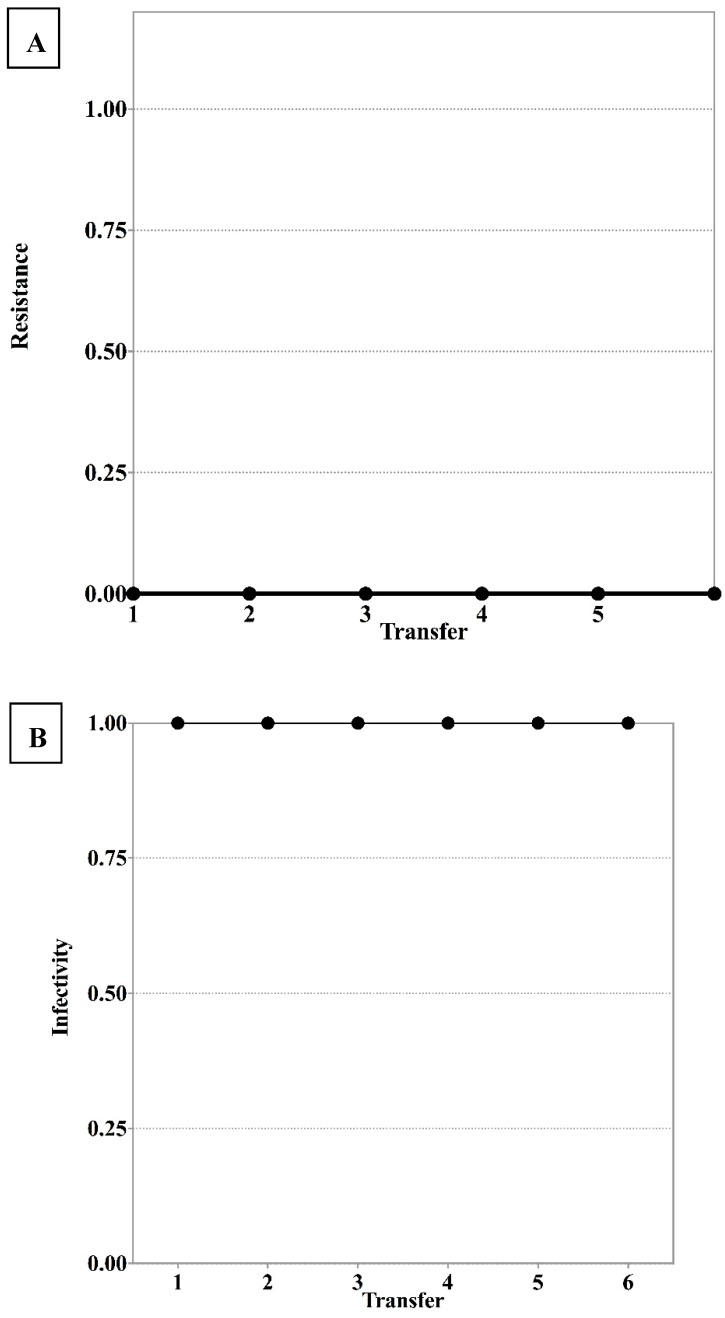
(**A**) Bacteria isolated from the control where only bacteria were allowed to evolve were shown to be susceptible to the ancestral phage. (**B**) Phages from each transfer from the control where only phages were allowed to evolve were shown to be infective against ancestral bacteria.

**Figure 4 viruses-11-00188-f004:**
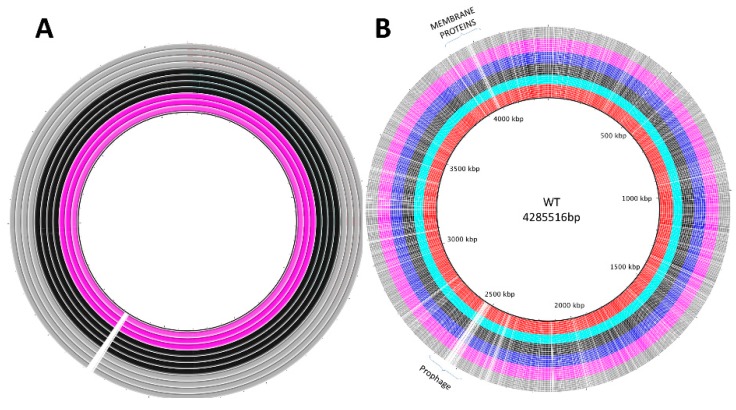
Visual comparison of genomic changes in the bacterial evolutionary control and phage resistant mutants over time. Inset (**A**) Bacterial control genomes; each ring represents a sequenced genome from the control, i.e. bacteria that evolve with no phage pressure. The single black line in the inner circle shows the ancestral genome. Color represents the transfer number in time: fuchsia rings correspond to transfer 1, black rings to transfer 3, and grey rings to transfer 6, white spaces represent mutations. Inset (**B**) shows de genomes of phage resistant mutants over time. The single black line in the inner circle shows the ancestral genome. Each ring represents a sequenced mutant genome and colors represent the transfer number in time: red rings correspond to transfer 1; turquoise to transfer 2; black to transfer 3; blue to transfer 4; fuchsia to transfer 5; and grey to transfer 6; white spaces represent mutations over time. Multiple point mutations at the prophage location (ca. 2500 bp) were found, in both control and phage resistant mutant bacteria.

**Figure 5 viruses-11-00188-f005:**
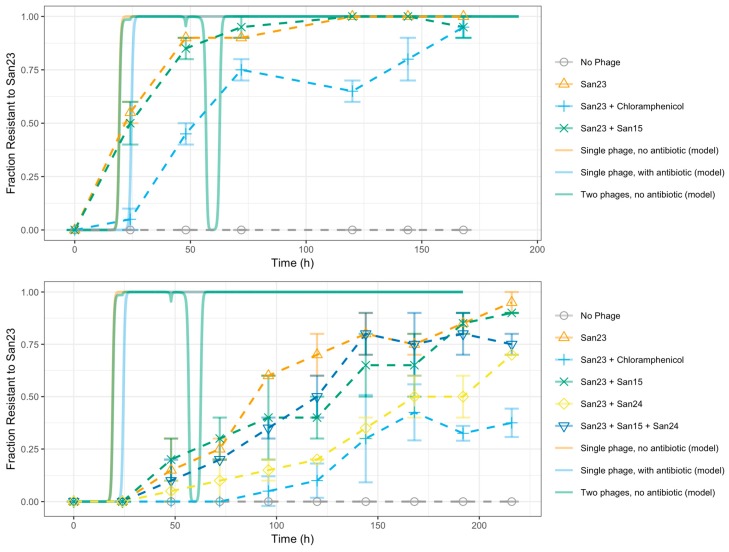
Increase in *S.* Enteritidis resistance to *φ*San23 through successive cultures. Each time series represents a different treatment of successively propagated *S.* Enteritidis cultures. Error bars portray standard error (*n* = 2). Dotted lines represent corresponding behavior as predicted by the mathematical model.

**Figure 6 viruses-11-00188-f006:**
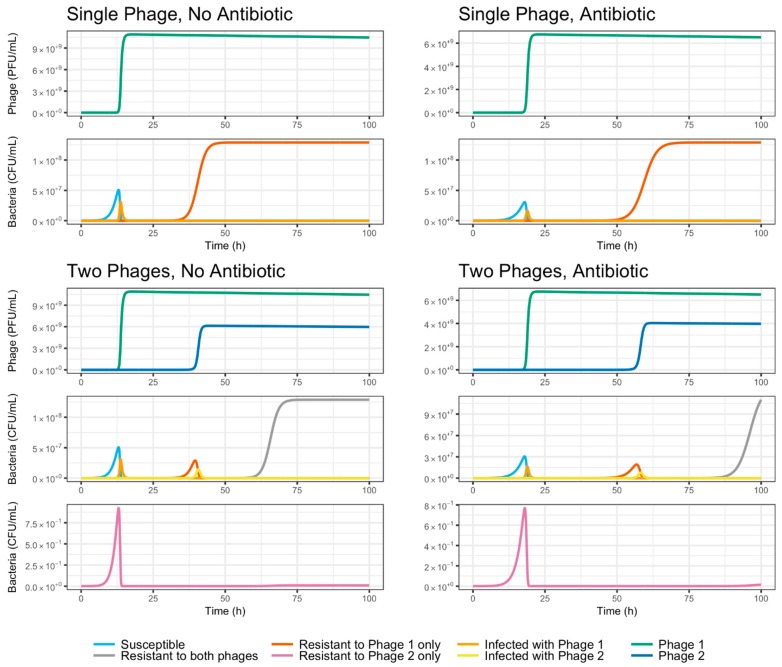
Population composition dynamics predicted under different treatment regimes (single phage without antibiotic, single phage with antibiotic, two phages without antibiotic, two phages with antibiotic). Dotted lines denote culture transfers, simulated by multiplying all compartments in the model by a dilution factor *v*. Note traces with similar behavior overlap each other near the horizontal axis, causing only the peaks of certain populations to be visible, as is the case of bacteria infected with Phage 1 in the two-phage systems (orange line), for example. Graphs have different vertical axis scales in order to better view dynamics.

**Figure 7 viruses-11-00188-f007:**
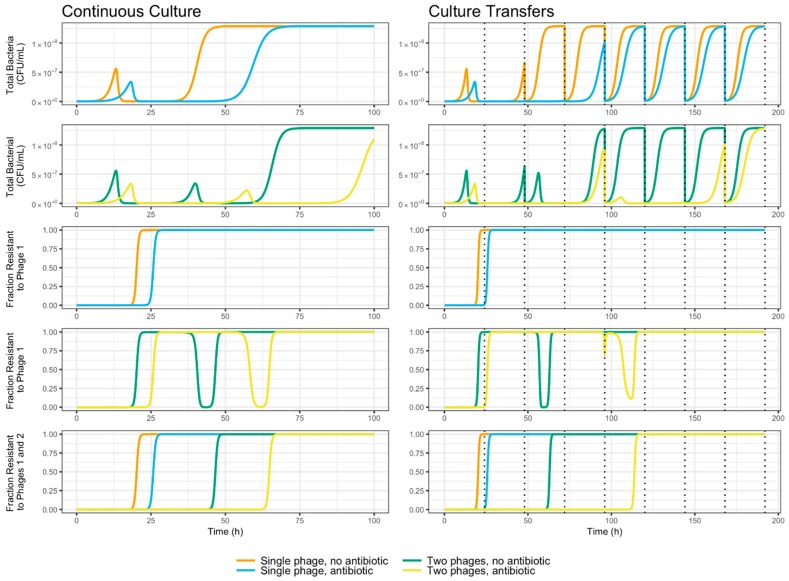
Bacterial population total size and resistance to phage predicted under different treatment regimes (single phage without antibiotic, single phage with antibiotic, two phages without antibiotic, two phages with antibiotic). Dotted lines denote culture transfers, simulated by multiplying all compartments in the model by a dilution factor *v*.

**Table 1 viruses-11-00188-t001:** Parameters used in ODE model for one- and two-phage infection of a bacterial population.

Constant	Symbol	Value	Units	Reference
Intrinsic growth rate of bacteria	*r*	0.68885	h^−1^	This study
Relative growth rate penalty of bacteria in presence of antibiotic	*α*	0.33736	--	This study
Presence/absence of antibiotic	*a*	{1,0}	--	--
Bacterial carrying capacity	*k*	1.2888 × 10^8^	CFU/mL	This study
Mutation probability	*m*	1.4000 × 10^−9^	--	This study, [71]
Adsorption rate for phage 1	*β* _1_	1.8000 × 10^−9^	mL pfu^−1^ h^−1^	*
Adsorption rate for phage 2	*Β* _2_	1.0000 × 10^−9^	mL pfu^−1^ h^−1^	*
Burst size for phage 1	*b_1_*	165	PFU/mL	*
Burst size for phage 2	*b_2_*	120	PFU/mL	*
Phage 1 virion decay rate	*µ*1	4.7917 × 10^−4^	h^−1^	[63,64,72]
Phage 2 virion decay rate	*µ*2	4.7917 × 10^−4^	h^−1^	[63,64,72]
Phage 1 latent period	*λ* _1_	0.5	h	*
Phage 2 latent period	*Λ* _2_	0.5	h	*
Inoculation volume as a fraction of the culture volume	*v*	6.000 × 10^−3^	--	--

*: Determined experimentally, [20].

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
