# Peer review of "Host Resistance, Genomics and Population Dynamics in a Salmonella Enteritidis and Phage System"

_viruses, 2019, doi:10.3390/v11020188_

Round 1

Reviewer 1 Report

The manuscript of “Host Resistance, Genomics and Population Dynamics 2 in a Salmonella Enteritidis and Phage System” by Angela Victoria Holguín et al, studied the population dynamics of Salmonella under different conditions. The results obtained are interesting and would be significant for developing phage therapy for treating multi-drug resistant bacterial infections. However, the presentations of the manuscript need to be improved to make it easier to understand.

Major points:

1.      How were the fraction resistant to San 23 (Figure 2 and 5) determined experimentally? This must be described clearly in the method.

2.      Figure 2: from the experiment description of S. Enteritidis and Phage φSan23 evolutionary dynamics, there should have both bacteria and phages from each transfer. This figure may need to separate into two figures so that one shows how bacteria obtained from each transfer will interact with phages from past, contemporary and future phages (plus control bacteria), and the other one shows how phages obtained from each transfer will interact with bacteria from past, contemporary and future bacteria (plus control phages)

3.      Both Figure 3 and 4: no much real information was shown except that they look fancy. I would like to suggest the authors should list key mutations in each transfer and how these mutations are changes along each transfer under different treatments.

4.      Figure 5: the presentation is bad since it could not see the differences between the lines from experiments and those from the mathematical models.

5.      In order to prove that the model developed is useful, additional experiments should be performed to test if development of resistant bacteria could be delayed or avoided under the best conditions (for example, under how many phages and antibiotics) predicted by the models.

Minor points: 

1.      Most of the introduction (from line 62 to 109) are discussing the resistance mechanism of bacteria to phages. But the resistance mechanisms are not the focus of the current study. It should summarize more previous studies on bacteria and phage evolutionary dynamics/models.

2.      In the discussion part, the authors should discuss what are the novelty of the current study compared to the previous studies, and what are the limitations of the models.

Author Response

January 27, 2019

REVIEWER 1

MAJOR POINTS

1.      How were the fraction resistant to San 23 (Figure 2 and 5) determined experimentally? This must be described clearly in the method.

Susceptibility to φSan23 was evaluated for ten colonies from each culture using a modified protocol of spot assays on double agar [21]. Briefly, 1 ml cultures of each colony were grown overnight in nutrient broth, after which 100 µl of culture were mixed with 100 µl of soft agar and plated onto nutrient agar. 3 µl of φSan23 suspension [10 × 109 PFU/ml] were deposited on the surface of each soft agar surface, and plates were inspected for lysis plaques after 24 h.

In order to clarify things, we could then add the following phrase at the end of that same paragraph: "The fraction resistant to φSan23 for each treatment was then calculated as the number of soft agar cultures with no lysis plaques divided by ten."

2.      Figure 2: from the experiment description of S. Enteritidis and Phage φSan23 evolutionary dynamics, there should have both bacteria and phages from each transfer. This figure may need to separate into two figures so that one shows how bacteria obtained from each transfer will interact with phages from past, contemporary and future phages (plus control bacteria), and the other one shows how phages obtained from each transfer will interact with bacteria from past, contemporary and future bacteria (plus control phages).

The phage and bacteria controls graphs were added to the manuscript. They specified that bacteria isolated from the control where only bacteria were allowed to evolve showed to be susceptible to the ancestral phage, and that phages from each transfer from the control where only phages were allowed to evolve showed to be infective against ancestral bacteria. In the case of the graph separating phage and bacteria, we consider that would not give extra information as the controls graphs would do, as the graph would be the opposite of the bacteria one. The proportion that is not showing resistant would be the proportion of infectivity.

3.      Both Figure 3 and 4: no much real information was shown except that they look fancy. I would like to suggest the authors should list key mutations in each transfer and how these mutations are changes along each transfer under different treatments.

We agree there is not much real information in the phylogenetic tree, we decided to move the graph to supplemental material.

4.      Figure 5: the presentation is bad since it could not see the differences between the lines from experiments and those from the mathematical models.

One possible solution might be switching the dotted and solid lines, as shown here:

It is important to note that the solid lines superimpose each other at different points.

5.      In order to prove that the model developed is useful, additional experiments should be performed to test if development of resistant bacteria could be delayed or avoided under the best conditions (for example, under how many phages and antibiotics) predicted by the models.

This is a good point, however, it is worth taking into account that to theoretically evaluate the effect of additional phages on the system, the structure of the model must change significantly. Adding a single phage implies in six additional compartments besides the eight illustrated in Figure 1, thus resulting in a 14-dimensional ordinary differential equation system which would complicate the analytical exploration shown. This seemed excessive given that the main point being made with the model is that the addition of a non-lethal growth-inhibiting antibiotic can amplify the efficacy of phage therapy. On the other hand, the experiments to test if development of resistant bacteria can ideally be delayed or avoided were indeed performed, and are shown in Figure 5. The addition of an antibiotic increases the time taken to evolve resistance against φSan23.

Minor points:

1.      Most of the introduction (from line 62 to 109) are discussing the resistance mechanism of bacteria to phages. But the resistance mechanisms are not the focus of the current study. It should summarize more previous studies on bacteria and phage evolutionary dynamics/models. We are summarizing the introduction.

2.      In the discussion part, the authors should discuss what are the novelty of the current study compared to the previous studies, and what are the limitations of the models.

The following paragraph is added to the manuscript:

“The current study showed the phage interaction of a resistant to antibiotics Salmonella Enteritidis and the dynamics under different treatments. The results determined that the phage-bacteria interaction change from system to system as well as this particular interaction did not demonstrate antagonist coevolution but mutational asymmetry. On the other hand, one the most important limitations of the study is that in the real world environment the conditions are different from the conditions we tested and the bacteria may not be that “comfortable” as it was under our conditions.”

Reviewer 2 Report

 This manuscript conducted co-evolution and population dynamtics evaluation test between one Salmonella Enteritidis (SE) strain and a set of phages (one phage, a combination of phages, and phages plus an antibiotic). The results and the findings are both comprehensive, important and interesting, which could be used to understand the interactions between phages and their targeting pathogen, also in potential phage-based therapy or pathogen control. However, before the manuscript can be officially accepted for publication, some of my comments and questions may need to be addressed first.

Major questions/comments:

1.     I noticed there were more than 18 strains at first, but only one was selected for test. Do you think it is possible that if changing to another S. Enteritidis strain, the results might be total different?

2.     The introduction part can be more concise if possible.

3.     Based on the phylogenetic tree, the transfers 1 to 6 mixed together without a clear generation-based evolution pattern. Why would this happen? Any explanation?

4.     Have you tried to build a phylogenetic tree just using the three cell wall or capsule genes? It might help to filter out noises.

5.     Have you considered the fact that in real world environment, SE might can’t have the “comfortable” environment as used here? This might be something you need to consider in your future research after this manuscript.

6.     For the dynamics model, based on Figure1 and your description, it seems one of the basic assumption for your mathematical model is the phages won’t interact with each other in resistance directly. Is it true? It seems from Figure5, the adding of another phage may lower the resistance of San23.

Other comments:

1.     Line 20: are there any metagenomes and related analysis in this project?

2.     Line 160: for SE and phage evolutionary dynamics, can you add a flowchart instead of a large paragraph to make it more reader friendly?

3.     Line 186: 60 to “Sixty”.

4.     Line 214: can you add the parameters of SNP calling?  Are the raw reads genomes also uploaded to NCBI?

5.     Line 215 to 217: doesn’t snp calling tell you the difference already? Why use mauve and brig to determine the differences? Or do you mean to give a global view for snp mutants viewing? A little bit confused.

6.     Line 221: the order of analysis should be changed in the manuscript. The paragraph between line 221 and line 236 should be moved forward before paragraph line 212 and 220.

7.     Line 226: for bowtie2, do you mean reference-based assembly?

8.     How do you select best assembly? use N50,or others? Can you list them in your manuscript?

9.     Line 227 to Line 230: Based on my experience, there are differences for the annotations, what would you select when you meet disagreement?

10.     Line 417 to Line 421: Can you list the SNP density among the three genes in somewhere of this manuscript?

11.     Line 421: Do u have a table to show which genomes in which transfer have snps in those three genes? And their location and whether the SNPs changed amino acids sequences or not?

12.     Line 421: Do the three genes located in the “Membrane proteins” region of Figure 3-B?If not, can you label the three genes in Figure 3-B?

13.     Line 432 to 433: interesting, do you mean the transfer-control genomes have snps on the prophage region too?

14.     For figure4, can you change the scale bar to SNP distance?

15.     For figure4, what are those numbers of each genome? It seems they stand for genomes of different transfers, but can you describe in the context?

16.     For figure4, all the genomes from different transfers mixed together. Have you did some analysis to see which SNPs only in transfer 1, 2, .. 6? This might help you find some evolution in a batch of noises.

17.     Line 477 to Line 478: can you explain to me the meaning of dotted lines? e.g. give an brief example here to make it more reader friendly?

Author Response

January 27, 2019

REVIEWER 2

Major questions/comments:

1.     I noticed there were more than 18 strains at first, but only one was selected for test. Do you think it is possible that if changing to another S. Enteritidis strain, the results might be total different?

It is difficult to say as we have noticed the interaction of bacteria-phage change according to the model. However, due the conditions and resistance of the bacteria strains I do not think the results would be super different. But, this is a personal perspective, it is necessary to determine that experimentally.

2.     The introduction part can be more concise if possible. We agree, we modified the introduction.

3.     Based on the phylogenetic tree, the transfers 1 to 6 mixed together without a clear generation-based evolution pattern. Why would this happen? Any explanation?.

This phylogenetic tree represents SNPs over 12 days we performed the experiment. I believe the bacteria does not show a mutational pattern to address the phage infection, but they mechanism is aleatory mutations that can be less energy consuming and can lead to a mutation on the membrane proteins blocking the phage-bacteria interaction. Mutational asymmetry.

4.     Have you tried to build a phylogenetic tree just using the three cell wall or capsule genes? It might help to filter out noises.

There is not much interesting information with this modifications, so we decided to bring the graph to supplemental material.

5.     Have you considered the fact that in real world environment, SE might can’t have the “comfortable” environment as used here? This might be something you need to consider in your future research after this manuscript.

Yes, I agree, and we are adding  a last paragraph about limitations of the models in the discussion.

6.     For the dynamics model, based on Figure1 and your description, it seems one of the basic assumption for your mathematical model is the phages won’t interact with each other in resistance directly. Is it true? It seems from Figure5, the adding of another phage may lower the resistance of San23.

This is are correct, the model assumes there is negligible cross-resistance between the phages studied. Section "Model Parameters and Assumptions" writes:

The model assumes the resistance mechanisms to each phage are independent of each other. In the case of φSan23, resistance is mediated by mutation of a surface receptor as was determined in this work; so far, we have no information about the other phages resistance mechanisms. Certain mechanisms of resistance, such as capsules or restriction enzymes, increase resistance to multiple phages [5]. This model does not take these mechanisms into account.

The phages selected were chosen for their distinct character in infection parameters. The experimental results presented in Figure 5 do in fact show an increase in the time taken to evolve resistance to φSan23 when in co-culture with additional phages. Nevertheless, this is explained by the fact that additional phages will kill bacteria resistant to φSan23 but susceptible to the other phages, resulting in a slower net growth of the φSan23-resistant population, a phenomenon illustrated by the mathematical model itself. This is the same dynamic seen in combination therapy strategies for infectious disease or cancer, in which the mechanisms of resistance for each treatment used are distinct from each other (thus having no interaction with each other in resistance directly), but result in a longer time to evolve resistance to any (and all) of the therapies themselves.

Other comments:

1.     Line 20: are there any metagenomes and related analysis in this project? It was a typo, we meant proteomics.

2.     Line 160: for SE and phage evolutionary dynamics, can you add a flowchart instead of a large paragraph to make it more reader friendly? Im adding a video that shows the mythology as supplemental material.

3.     Line 186: 60 to “Sixty”. Done

4.     Line 214: can you add the parameters of SNP calling?  Are the raw reads genomes also uploaded to NCBI? I added.

5.     Line 215 to 217: doesn’t snp calling tell you the difference already? Why use mauve and brig to determine the differences? Or do you mean to give a global view for snp mutants viewing? A little bit confused. Yes, that’s right. We wanted to give a global view por snp mutants. We did not feel the snp calling tree was showing where most of the mutations were localize.

6.     Line 221: the order of analysis should be changed in the manuscript. The paragraph between line 221 and line 236 should be moved forward before paragraph line 212 and 220.

7.     Line 226: for bowtie2, do you mean reference-based assembly? Yes, I do

8.     How do you select best assembly? use N50,or others? Can you list them in your manuscript? sure

9.    Line 227 to Line 230: Based on my experience, there are differences for the annotations, what would you select when you meet disagreement?

10.     Line 417 to Line 421: Can you list the SNP density among the three genes in somewhere of this manuscript? Yes.

11.     Line 421: Do u have a table to show which genomes in which transfer have snps in those three genes? And their location and whether the SNPs changed amino acids sequences or not? Yes, Im adding as supplemental Figure S6.

12.     Line 421: Do the three genes located in the “Membrane proteins” region of Figure 3-B?If not, can you label the three genes in Figure 3-B? Yes, they located there.

13.     Line 432 to 433: interesting, do you mean the transfer-control genomes have snps on the prophage region too? Yes!!!

14.     For figure4, can you change the scale bar to SNP distance?. This figure was moved to supplemental material.

15.     For figure4, what are those numbers of each genome? It seems they stand for genomes of different transfers, but can you describe in the context? Yes, they are basically the nomenclature we used to determine the relicate and the transfer.

16.     For figure4, all the genomes from different transfers mixed together. Have you did some analysis to see which SNPs only in transfer 1, 2, .. 6? This might help you find some evolution in a batch of noises. I tried to summarize that in the figure S6.

17.     Line 477 to Line 478: can you explain to me the meaning of dotted lines? e.g. give an brief example here to make it more reader friendly? The caption currently reads:

Dotted lines represent corresponding behavior as predicted by the mathematical model.

To clarify, perhaps we can write “Dotted lines represent the fraction of resistance for each of the strains as predicted by the mathematical model (e.g., green showing two phages without antibiotics).”

Round 2

Reviewer 1 Report

Necessay corrections have been made. I have no more suggestions.

Author Response

Dear reviewer, 

Thank you so much

Reviewer 2 Report

I think the authors addressed most of my comments and I agree that it's a qualified manuscript to be accepted. But if possible, I would push the authors to polish some parts additionally, for example, 

1) if bowtie2 used for reference based assembly, then the N50 would be super high if you use a complete genome as reference. How would you select the assembly using N50 (line 212)? It's not a fair comparison frankly speaking. Also the first half of line 212 is redundant since there was sentences describing assembly already above.

2) can you explain a little bit why the transfer-control genomes have snps on the prophage region too? are they mutation hot spots?

Again, it is the authors' call to address them or not, and it is also can be accepted if the authors don't want to change. 

Author Response

Dear reviewer, 

Thank you for comments, here are the modifications.

1) if bowtie2 used for reference based assembly, then the N50 would be super high if you use a complete genome as reference. How would you select the assembly using N50 (line 212)? It's not a fair comparison frankly speaking. Also the first half of line 212 is redundant since there was sentences describing assembly already above.

I agree, I need to explain better that paragraph as we performed first the assembly for the ancestral bacteria and then for the mutants. For the ancestral bacteria we used SPAdes and Assembler 1.0 and we selected the best assembly based on N50. Then We used the ancestral bacteria to assembly mutants using Bowtie2, in parallel we perfom the de novo assembly with SPAdes and Assembler 1.0 in order to compare the assembly between them. I added this paragraph:

 "The ancestral bacterial assembly was carried first using SPAdes and Assembler 1.0, we selected best assembly based on N50, then the mutants genome assembly was performed with Bowtie2, SPAdes and Assembler 1.0."

For the line 212 we talked about the genomics perform with phages, thats why may sound a bit redundant. 

2) can you explain a little bit why the transfer-control genomes have snps on the prophage region too? are they mutation hot spots? Sure,

Prophages have a very important role on bacteria, in this case in Salmonella sp. gene mobility, adaptation, differentiation. For instance, there was found that SNPs in prophages of Listeria monocytogenes differentiate clones from epidemic and outbreak. So, from my point of view, this prophage may be a mutational hot spot. However, I also believe there is room for more investigation on whats happening with the prophage over time. 
